# Characterization of arbovirus infections in patients within Haiti by screening discarded malaria rapid diagnostic test cassettes

Graham A. Matulis[1,2,3,4,5], Haley P. Smith[4,6], Grant Hall[7], Rachel S. Katich[4], Korey L. Delp[6], Christina E. Douglas[4], Jennifer Williams[6,7], Shawn Hirsch[7], Raina Kumar[7], Ian Pshea-Smith[1,8], Abigail A. Lilak[1,2,3], Bernard A. Okech[9], Keersten Ricks[4], Christopher P. Stefan[4], Alexandre Existe[10], Jeffrey R. Kugelman[7], Ian Sutherland[1,11], James Dunford[1,12], Jacques Boncy[10], Michael E. von Fricken [1,2☯*], Jeffrey W. Koehler[1,4☯*]

1 One Health Center of Excellence, College of Public Health and Health Professions, University of Florida, Gainesville, Florida, United States of America, 2 Department of Environmental and Global Health, University of Florida, Gainesville, Florida, United States of America, 3 Emerging Pathogens Institute, University of Florida, Gainesville, Florida, United States of America, 4 Diagnostic Systems Division, US Army Medical Research Institute of Infectious Diseases, Frederick, Maryland, United States of America, 5 Oak Ridge Institute for Science and Education, US Army Medical Research Institute of Infectious Diseases, Diagnostic Systems Division, Frederick, Maryland, United States of America, 6 Chenega Professional and Technical Service, US Army Medical Research Institute of Infectious Diseases, Frederick, Maryland, United States of America, 7 Molecular Biology Division, US Army Medical Research Institute of Infectious Diseases, Frederick, Maryland, United States of America, 8 Spatial Epidemiology & Ecology Research Laboratory, Department of Geography, University of Florida, Gainesville, Florida, United States of America, 9 Department of Preventative Medicine and Biostatistics, Uniformed Services University, Bethesda, Maryland, United States of America, 10 Laboratoire National de la Sante Publique, Ministere de la Sant e Publique et de la Population, Port-au-Prince, Haiti, 11 Purdue University Global, West Lafayette, Indiana, United States of America, 12 Lee County Mosquito & Hyacinth Control Districts, Lehigh Acres, Florida, United States of America

☯ These authors contributed equally to this work.
* mvonf@ufl.edu (MEvF); jeffrey.w.koehler4.civ@health.mil (JWK)

## Abstract

### Background

Arboviruses represent a diverse group of pathogens transmitted by arthropod vectors. Within Haiti, arboviruses responsible for previous outbreaks include dengue virus, Zika virus, and chikungunya virus. Recent security concerns within Haiti have interrupted broader surveillance efforts, creating challenges for public health agencies tasked with monitoring for vector-borne diseases. In this study, we aimed to better understand circulating arboviruses within Haiti using nucleic acids stored in discarded malaria rapid diagnostic tests (RDTs).

### Methodology/principal findings

RDTs were collected between 2021–2023 from febrile patients presenting to health care centers throughout the Sud and Ouest Departments of Haiti. Test strips were removed from the RDT cassettes, and total nucleic acid was extracted. Samples

**Data availability statement:** Sequence data is publicly available on GenBank (Accession numbers PX398242- PX398251). The testing results for all samples are provided in S1 Fig.

**Funding:** This work was funded by the Armed Forces Health Surveillance Branch (AFHSB), Global Emerging Infections Surveillance (GEIS) Section, under ProMIS ID P0129-20-RD-114361, P0118-24-RD, and P0154_24_EC. This research was supported in part by an appointment to the Department of Defense (DOD) Research Participation Program administered by the Oak Ridge Institute for Science and Education (ORISE) through an interagency agreement between the U.S. Department of Energy (DOE) and the DOD. ORISE is managed by ORAU under DOE contract number DE-SC0014664. The funders had no role in study design, data collection and analysis, decision to publish, or preparation of the manuscript.

**Competing interests:** The authors have declared that no competing interests exist.

were initially tested for sample integrity using a human RNase P real-time PCR assay, followed by a flavivirus spp. real-time PCR assay. A total of 52 RDTs tested positive by the flavivirus spp. assay, and an additional 21 were indeterminant. Testing all flavivirus spp. positive and indeterminant samples with a DENV quadraplex assay resulted in 68 samples testing positive for DENV-3. All samples testing positive for DENV-3 were collected in 2023. NGS sequencing and subsequent phylogenetic analysis demonstrated high sequence similarity to sequences published from the Caribbean region between 2022–2023. A subset of the flavivirus negative RDTs were tested using alphavirus spp. (n = 517) and Oropouche virus (n = 293) real-time RT-PCR assays. No samples tested positive using either the alphavirus spp. (0/517) or Oropouche virus (0/293) assays.

## Conclusions/significance

These results demonstrate the context-specific utility of discarded malaria RDTs for remote arbovirus surveillance among febrile patients, with potential for viral characterization. The exclusive finding of DENV-3 within these samples is concordant with the DENV-3 outbreak that was observed throughout the Americas in 2023. As political insecurity continues within Haiti, malarial RDTs represent an important tool for high level surveillance of novel public health threats.

### Author summary

Public health activities such as vector-borne disease surveillance are largely interrupted within countries experiencing humanitarian crises. Within Haiti, a country currently experiencing ongoing political instability, numerous vector-borne viruses (arboviruses) with significant impacts to population health have been detected. Such scenarios necessitate new methods for continued disease surveillance to detect and respond to emerging public health threats. In this study, we demonstrated the repurposed utility of rapid diagnostic tests (RDTs), used in-country for malaria diagnosis among febrile illness patients, for molecular-based surveillance of circulating arboviruses. By extracting genetic material present in the patient blood sample stored within the RDT strip, we detected dengue virus serotype 3 and genetically characterize multiple samples with sequencing. This study presents an extended utility of RDTs for remote molecular-based surveillance of infectious diseases. While this approach requires both coordination with local health care systems to collect anonymized RDTs and access to appropriate biosafety infrastructure for sample processing, such methodology can be used to enhance disease surveillance in other countries undergoing similar crises or in locations where circulating infectious disease profiles have been largely uncharacterized due to logistics-related lack of surveillance capacity.

## Introduction

As of 2023, 537 different arboviruses were recognized globally, with more than 130 of these considered capable of causing disease in humans [1]. Infections with arboviruses can vary in clinical severity, from completely asymptomatic to encephalitis and hemorrhagic fever [2]. Arboviruses are estimated to cause up to 700,000 deaths annually, including 22,000 deaths from dengue virus (DENV) and 30,000 deaths from yellow fever virus (YFV) [3,4]. Moreover, at least 3.9 billion people currently live within regions that place them at risk for mosquito-borne arboviral infections. Human activities and global climatic change are expected to increase the burden of arboviruses through the expansion of vector species distribution patterns, with some models projecting an increase of 4.7 billion people at risk by 2070 compared to the population at risk between 1970–1999 [5,6].

In Haiti, multiple arboviruses have previously been reported, including chikungunya virus (CHIKV), DENV, Maradiaga virus (MADV), Mayaro virus (MAYV), Melao virus (MELV), Oropouche virus (OROV) and Zika virus (ZIKV) [7–9]. While DENV circulates endemically within Haiti, these other viruses have been detected in either limited instances or during outbreak circumstances. In April 2014, an outbreak of CHIKV was reported in Haiti, amounting to around 65,000 suspected cases, with no serological evidence of CHIKV presence prior to 2014 [10,11]. Between October 2015-September 2016, the Haitian Ministry of Health reported 3,036 cases of ZIKV during the ZIKV epidemic in the Americas, although ZIKV had been present within the Haitian population as early as December 2014 [12,13]. The Pan-American Health Organization (PAHO) has not reported CHIKV or ZIKV in Haiti since 2019. However, a recent study of school children sampled during 2021 reported a few instances of CHIKV and ZIKV RNA in sera, suggesting a continued transmission of these arboviruses within Haiti [14].

Given the diversity of arboviruses that are found within Haiti, symptom-based diagnosis of patients presenting with acute febrile illness is challenging, necessitating the use of assay-based diagnostics [14]. While antibody- and antigen-based rapid diagnostic tests (RDTs) have been commercially developed for arboviruses such as DENV, CHIKV, and ZIKV, these RDTs report challenges regarding assay specificity and sensitivity, impacting their utility within regions where multiple arboviruses may co-circulate [15–17]. Due to the frequency of arbovirus emergence and re-emergence events in Haiti, continued molecular arboviral monitoring is necessary to ensure accurate and prompt responses in the instance of an outbreak. This is especially relevant given significant barriers to healthcare access amidst the current humanitarian crisis. Genetic material stored within used RDTs, hold the potential for remote molecular analyses in areas with security concerns, allowing for arbovirus monitoring to continue despite ongoing political instability within Haiti. The utility of used arboviral RDTs for molecular studies has been previously demonstrated for DENV, CHIKV, and ZIKV [18–21]. Here we describe the use of discarded malarial RDTs from febrile patients to assess the presence of circulating arboviruses within Haiti.

## Methods

### Ethics statement

Research involving human subjects adhered to the principles identified in the Belmont Report (1979) and, unless classified as exempt, was conducted in accordance with an IRB-approved protocol and in compliance with DoD, federal, and state statutes and regulations relating to the protection of human subjects. The George Mason University Institutional Review Board (IRB) determined (OSP #12153A) that their role does not meet the definition of human subject research under the purview of the IRB according to federal regulations. The US Army Medical Research Institute of Infectious Diseases component was certified as exempt (FY24–15) and was conducted in accordance with the conditions specified in connection with an Exemption Certificate.

### Study sites and samples

A total of 1,886 malaria HRP2-based rapid diagnostic tests, (SD Bioline and First Response) were collected in Haiti within the Sud and Ouest departments between 2021 and 2023. These RDTs are a subset of a larger malaria RDT sample set

(n = 2,073), with the RDTs not included in this study representing samples dedicated for genetic characterization of *P. falciparum* within Haiti. The RDTs were from patients presenting to local health care centers with febrile illness. The RDTs were anonymous, and the only information associated with the RDT is month of collection, test number, and clinic site. Those who participated in the sample processing and analyses were not provided with any metadata regarding the RDTs, and samples were given a new sample number as they were processed.

RDT samples were collected and stored at in a Ziplock bag room temperature at the target health care centers within Haiti. On a monthly basis, samples were transferred to the Laboratoire National de la Sante Publique where they were stored at 4°C up to two years until transfer to the US Army Medical Research Institute of Infectious Diseases (USAMRIID). Upon arrival at USAMRIID, RDTs were stored at room temperature (20–22°C) until they were processed.

### RDT processing and sample extraction

All RDT sample processing was conducted at biosafety level 2 (BSL-2) laboratory conditions. Standard infectious disease precautions were taken including a biosafety cabinet and personal protective equipment. After opening RDT cassettes, sample strips were removed and cut into pieces for processing. Briefly, cut strips were placed into a tube containing 500 µL of ATL buffer (Qiagen, Hilden, Germany) and incubated at 56°C for 15 minutes. Strips were then removed from the tubes, followed by extraction of total nucleic acid (TNA) from the remaining supernatant using the EZ1 DSP Virus Kit (Qiagen, Hilden, Germany) and the EZ1 Advanced XL (Qiagen, Hilden, Germany) according to the manufacturer's instructions. Extractions were eluted into 60 µL and stored at -70°C until analyzed. TNA quality was assessed using a real-time RT-PCR assay to detect the endogenous human gene *RNase P* assay. Samples were run in singlet using TaqPath 1-Step RT-qPCR Master Mix, CG (ThermoFisher, Waltham, Massachusetts, USA), RNase P Master Mix (Cat# 10006836, #10006837, #10006838, IDTDNA, San Diego, California, USA), and molecular grade water (Ambion, Austin, Texas, USA), for a final volume of 20 µL. PCR cycle conditions were 25°C for 2 minutes, 50°C for 15 minutes, 95°C for 2 minutes, and 45 cycles of 95°C for 3 seconds and 55°C for 30 seconds. A sample was considered positive for RNase P with a Cq value of <40 cycles and an appropriate, sigmoidal PCR curve.

### Flavivirus detection by real-time RT-PCR

Samples were tested for flaviviruses using a real-time RT-PCR assay modified from Patel and colleagues targeting a conserved region of the flavivirus *nsp5* gene [22]. The same primers and probes were used with cycling conditions adapted for use with the 4X TaqPath 1-Step RT-qPCR Master Mix, CG (ThermoFisher, Waltham, Massachusetts, USA). Samples were run in singlet with the *nsp5* primers and probes and molecular grade water (Ambion, Austin, Texas, USA) for a final reaction volume of 20 µL. PCR cycling conditions were 25°C for 2 minutes, 50°C for 15 minutes, 95°C for 2 minutes, and 45 cycles of 95°C for 3 seconds and 55°C for 30 seconds. Samples were considered positive with Cq values <40 and an appropriate, sigmoidal PCR curve. A sample was considered indeterminate when a sigmoidal PCR curve was produced but the Ct value was > 40.

### Flavivirus speciation by Sanger sequencing

Sanger sequencing of selected flavivirus-positive samples was conducted to identify the specific flavivirus species. The flavivirus *nsp5* gene was amplified using the primers from the flavivirus spp. real-time RT-PCR assay [22] and the SuperScript III RT/Platinum Taq Mix kit (ThermoFisher, Waltham, Massachusetts, USA). The PCR cycling conditions were 50°C for 30 minutes, 94°C for 2 minutes, and 50 cycles of 94°C for 15 seconds, 60°C for 30 seconds and 70°C for 1 minute. PCR products were cleaned using the ExoSAP-IT Express PCR Cleanup kit (ThermoFisher, Waltham, Massachusetts, USA) following the manufacturer's protocol. Cycle sequencing was conducted using the BigDye Terminator v3.1 Cycle Sequencing kit (ThermoFisher, Waltham, Massachusetts, USA) following

the manufacturer's protocol and using the same *nsp5* primers. The cycle sequencing PCR product was processed using the BigDye XTerminator kit (ThermoFisher, Waltham, Massachusetts, USA) and sequenced on the SeqStudio Genetic Analyzer (ThermoFisher, Waltham, Massachusetts, USA). Sequences were analyzed using CLC Genomics Workbench (Qiagen, Hilden, Germany) and BLASTn.

## DENV quadraplex real-time RT-PCR

Based on the Sanger sequencing results, all flavivirus positive samples were further tested using a multiplexed DENV serotype-specific real-time RT-PCR assay adapted from Santiago and colleagues [23]. Samples were run in singlet using the SuperScript III RT/Platinum Taq Mix kit (ThermoFisher, Waltham, Massachusetts, USA). Cycling conditions were 25°C for 1 min; 50°C for 15 min; 95°C for 2 min; 45 cycles of 95°C for 3 sec, 60°C for 30 sec. A sample was considered positive with a Cq value of <40 cycles and an appropriate, sigmoidal PCR curve. A sample was considered indeterminate when a sigmoidal PCR curve was produced but the Ct value was > 40.

## Whole genome sequencing of DENV-positive samples

A subset (n = 11) of DENV-positive samples were selected for further genomic characterization using next generation whole genome sequencing. Samples were chosen to represent each of the health care centers from which a positive sample was detected. Samples were enriched in single-plex using the Viral Surveillance Panel 2.0 (Illumina, San Diego, California, USA) and RNA Prep with Enrichment (Illumina, San Diego, California, USA). Samples were hybridized overnight and run 2x150 cycles on a MiSeq (Illumina, San Diego, California, USA) using a MiSeq V3 300 Cycle Reagent Kit (Illumina, San Diego, California, USA).

Sequencing reads were analyzed using an in-house pipeline. Briefly, paired reads adapters were trimmed using cutadapt v2.6. Reads were then aligned to the NCBI DENV-3 reference NC_001475.2 using bwa-mem (v 0.7.12). PCR duplicates were removed using Picard v 2.20.7 and reads with a mapping quality less than 20 were removed using samtools v1.13. Variants were called using bcftools v1.13 ignoring bases with a phred score less than 20 and filtered for a Qual >20 and depth >20. Samtools was used to create an alternative consensus sequence from the variant file with a minimum depth requirement of 20 to make a base call.

Phylogenetic trees were built with CLC Genomics Workbench using Neighbor Joining, Jukes-Canto method (1000 bootstrap replicates). DENV sequences were selected from NCBI virus having genomes >1000 bp. A randomized subset of 20 or fewer genomes stratified by country were downloaded for phylogenetic tree building. In total 690, 699, and 442 genomes for DENV-1, DENV-2, and DENV-3, respectively, were used.

## Alphavirus detection by real-time RT-PCR

A subset (n = 517) of flavivirus-negative samples were tested using an in-house real-time RT-PCR targeting a conserved region of the *nsP4* gene of alphavirus species. The primers included alpha-nsP4-F1047 (5'-TGAYATGTCNGCNGAR GAYTT) and alpha-nsP4-R1415a (5'- ATGTTGTMGTCGCCDATRAA) with the probe alpha-nsP4-p1288 (6FAM- ATGAT GAARTCNGGNATGTT-MGBNFQ). The assays were tested with a range of different alphaviruses, showing positive detection of Barmah Forest virus, Chikungunya virus, Mucambo virus, Venezuelan equine encephalitis virus, Getah virus, Eastern equine encephalitis virus, and O'nyong-nyong virus. Samples were chosen randomly with a prioritization of RDT samples from 2023 given concerns for viral RNA stability from older samples. Five microliters of TNA were combined with a master mix made up of 4X TaqPATH 1-Step RT-qPCR Master Mix, CG (ThermoFisher, Waltham, Massachusetts, USA), an in-house primer and probes mix, and molecular grade water (Ambion, Austin, Texas, USA), for a final PCR reaction volume of 20 µL. The PCR cycle conditions were 25°C for 2 minutes, 50°C for 15 minutes, 95°C for 2 minutes, and 45 cycles of 95°C for 3 seconds, 50°C for 30 seconds, and 60°C for 30 seconds. A sample was considered positive with a Cq value

of <40 cycles and an appropriate, sigmoidal PCR curve. A sample was considered indeterminate when a sigmoidal PCR curve was produced but the Ct value was > 40.

### Oropouche virus detection by real-time RT-PCR

A subset (n = 293) of flavivirus negative samples were run on an in-house real-time RT-PCR targeting the L segment RNA-dependent RNA polymerase gene of the OROV. The primers included OROV-L-F106 (5'-AGCRAAGGATATATGG CGHGATCT) and OROV-L-R183 (5'-AGRTTWGCAGCTCTGCAAAATTC) with the probe OROV-L-p136 (6FAM-TGATC GACAYAAYTAC-MGBNFQ). Samples were tested, showing positive detection of OROV. Samples were chosen randomly with a prioritization of RDT samples from 2023 given concerns for viral RNA stability from older samples. Five microliters of TNA were combined with a master mix made up of 4X TaqPATH 1-Step RT-qPCR Master Mix (ThermoFisher, Waltham, Massachusetts, USA), an in-house primer and probes mix, and molecular grade water (Ambion, Austin, Texas, USA), for a final PCR reaction volume of 20 µL. The PCR cycle conditions were 25°C for 2 minutes, 50°C for 15 minutes, 95°C for 2 minutes, and 45 cycles of 95°C for 3 seconds, 50°C for 30 seconds, and 60°C for 30 seconds. A sample was considered positive with a Cq value of <40 cycles and an appropriate, sigmoidal curve. A sample was considered indeterminate when a sigmoidal PCR curve was produced but the Ct value was > 40.

### Data visualization

Geographic visualizations of sampled healthcare center sites and pathogen-positive sites were created using ArcGIS Pro (Esri, 3.1.0).

## Results

### RDT processing

A total of 1,886 RDTs were collected in Haiti between 2021 and 2023. Of these, 72 samples were RDT-positive for *Plasmodium*. We initially sought to optimize nucleic acid extraction from the stored RDTs for RNA testing. Groups have previously demonstrated the use of RDT test strips to molecularly detect viral pathogens such as arboviruses [18–20]. Vongsouvath and colleagues found the highest viral RNA recovery was seen by processing the sample pad section of the used RDT [19]. Considering this finding, we utilized the entire RDT test strip to maximize nucleic acid recovery, processing a total of 1,884 RDTs. All samples included in this study were positive for the human endogenous gene RNase P, indicating that the TNA product was of sufficient quality to allow for human genes to be detectable. The overall average RNase P Cq values was 27.61 ± 3.53. The RNase P Cq values decreased over time from RDTs collected in 2021: 29.28 ± 2.89 (n = 573); 2022: 27.51 ± 2.42 (n = 44); and 2023: 24.85 ± 2.96 (n = 1,267). These data suggest a decrease in sample stability of about 1 log over the three-year collection period.

### Identification and analysis of arbovirus-positive samples

Total nucleic acid from RNase P-positive samples (n = 1,886) screened using a flavivirus real-time RT-PCR assay identified 52 positive, 21 indeterminate, and 1,811 negative samples. Flavivirus-positive samples were only detected in June-September 2023 samples from Chantal, Cayes, Port Salut, St. Louis du Sud, Les Anglais, and Port-a-Piment (Fig 1). To gain insight into which flaviviruses were being detected by the flavivirus spp. assay, *nsp5* amplicon Sanger sequencing was conducted on a subset of flavivirus spp. positive samples (n = 20), identifying DENV as the only flavivirus species present within the samples. Given this information, all flavivirus positive and indeterminate samples (n = 73) were tested using the DENV quadraplex assay, identifying 68 samples positive for DENV-3 (Table 1). Among the samples testing positive for DENV-3, two were from samples that were RDT positive for malaria. These co-infections were from Les Anglais (n = 2), Sud department. Attempts were made to identify the flavivirus present within the five flavivirus-positive and

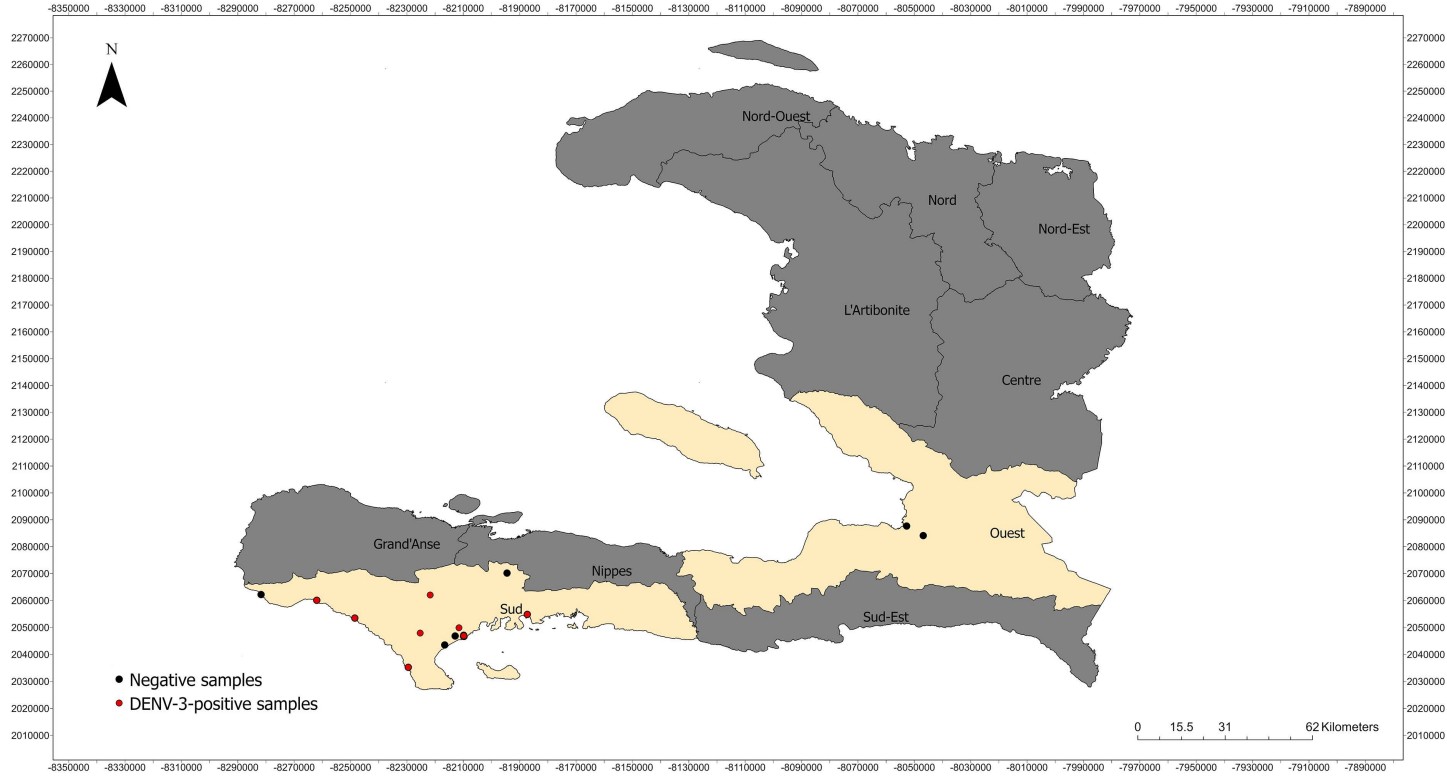

**Fig 1. Location of RDT samples and DENV-3-positive RDT samples.** A total of 1,884 RDTs were collected across southern Haiti between 2021 and 2023. Sampled departments are shaded beige while departments not sampled are shaded grey. Locations with no DENV-3 detected are shown as a black circle while locations positive for DENV-3 are shown as yellow boxes. The shapefile for the base layer of this map was obtained from https://geo-data.ucdavis.edu/gadm/gadm4.1/shp/gadm41_HTI_shp.zip. License information available at https://gadm.org/license.html.

indeterminant samples that were negative for DENV through Sanger sequencing, although no flavivirus was identified from this analysis.

Given the presence of other arboviruses in the region, including the recent Oropouche virus outbreak in the Americas, we decided to test a subset of the samples for alphaviruses and Oropouche virus. None of the subset of samples tested positive for either an alphavirus species (0/517) or Oropouche virus (0/293).

## NGS sequencing and phylogenetic analysis

Next-generation sequencing of a subset (n = 11) of DENV-3 positive-samples from all sampled locations with positive samples resulted in ten full length and one partial length sequence for DENV-3. BLAST analysis of these genomes found significant homology to other DENV-3 isolates circulating in the Caribbean in 2022 and 2023 (Fig 2).

These reference sequences included clinical samples from Florida of recent visitors to Haiti and Cuba (PP709295.1, OR771141.1, OQ821539.1, OQ821558.1), and isolates from French Guiana (PP582683.1, PP582658.1, PP582639.1), Brazil (OQ706226.1), and Peru (PQ467858.1). Over 99.5% identity was observed between the sequences from this study and these reference sequences.

## Discussion

In this study, we demonstrated evidence for circulation of DENV-3 within the Sud Department of Haiti during June-September 2023, with an overall 2023 sample positivity rate of 5.36% among patients presenting to local clinics with

**Table 1. RDT samples analyzed by location and year. The number of flavivirus-positive samples and DENV positive samples are shown for each location.**

| Year | Location (Department) | # RDTs | flavivirus-positive/IND | DENV multiplex |
|---|---|---|---|---|
| 2021 | Bonne Fin (Sud) | 68 | 0 | – |
| | CS Collette (Sud) | 10 | 0 | – |
| | CityMed Cayes (Sud) | 173 | 0 | – |
| | CityMed Petion-Ville (Ouest) | 18 | 0 | – |
| | CS FINCA (Sud) | 35 | 0 | – |
| | CS Port-a-Piment (Sud) | 67 | 0 | – |
| | CS Saint Louis du Sud (Sud) | 8 | 0 | – |
| | CS Torbeck (Sud) | 66 | 0 | – |
| | Hopital FCS de P-au-P | 9 | 0 | – |
| | Port-Salut (Sud) | 97 | 0 | – |
| | Tiburon (Sud) | 22 | 0 | – |
| 2021/2022 | Hopital Regional Cayes (Sud) | 43 | 0 | – |
| 2022 | Hopital de Port Salut (Sud) | 1 | 0 | – |
| 2023 | Dispensarie de Chantal (Sud) | 15 | 3 | DENV-3: 3(20.00%) |
| | HIC Cayes (Sud) | 71 | 0 | – |
| | HIC Cayes – Batch 3 (Sud) | 78 | 4 | DENV-3: 4(5.13%) |
| | Hopital de Port Salut (Sud) | 215 | 9 | DENV-3: 9(4.18%) |
| | Hopital de St. Louis du Sud (Sud) | 73 | 5 | DENV-3: 4(5.48%) |
| | Hopital de Tiburon (Sud) | 98 | 0 | – |
| | Hopital Les Anglais (Sud) | 52 | 0 | – |
| | Hopital Les Anglais – Batch 3 (Sud) | 303 | 20 | DENV-3: 19(6.27%) |
| | Hopital OFATMA Sud (Sud) | 20 | 3 | DENV-3: 2(10.00%) |
| | Hopital Port-a-Piment – Batch 2 (Sud) | 37 | 0 | – |
| | Hopital Port-a-Piment – Batch 3 (Sud) | 197 | 25 | DENV-3: 23(11.67%) |
| | Hopital Port a Piment Public (Sud) | 41 | 0 | – |
| | Hopital Ste Anne (Sud) | 26 | 4 | DENV-3: 4(15.38%) |
| | MSF Port a Piment (Sud) | 43 | 0 | – |
| **Total** | | 1,886 | 73 | DENV-3: 68(3.60%) |

febrile illness. Considering individual sampling locations, positivity rates ranged from 4.2-20%, with Chantal, Hopital Ste Anne, and Port-a-Piment reporting the highest positivity rates. Two instances of DENV-3 co-infections with *Plasmodium falciparum* were detected in this study, bothsamples coming from Les Anglais. Phylogenetic analyses of sequences obtained through NGS demonstrated high sequence similarity to samples obtained from patients in the US who recently traveled to Haiti and Cuba in 2023 as well as clinical isolates from French Guiana and Brazil.

The profile of circulating arboviruses within Haiti has fluctuated over the past decade. Schoolchildren presenting to school clinics within the Ouest Department of Haiti between 2014–2016 have demonstrated evidence of infections with CHIKV, DENV-1, DENV-2, DENV-3, DENV-4, MADV, MAYV, MELV, OROV, and ZIKV [7–9,24,13]. More recently, a study of school children within the Ouest Department in 2021 reported the presence of CHIKV, ZIKV, and DENV-2 in serum [14]. The current study reports a change in the profile of circulating arboviruses within Haiti since 2021, with evidence of DENV-3 replacing DENV-2 as the circulating serotype within the country. The detection of DENV-3 within Haiti does align with PAHO reports of increased DENV-3 circulation within the Americas beginning in 2023, with phylogenetic analyses indicating that the introduction point of this widespread DENV-3 was within the Caribbean [25]. The lack of DENV in the 617 samples screened from 2021 and 2022 was unexpected and is potentially a result of degraded sample quality from

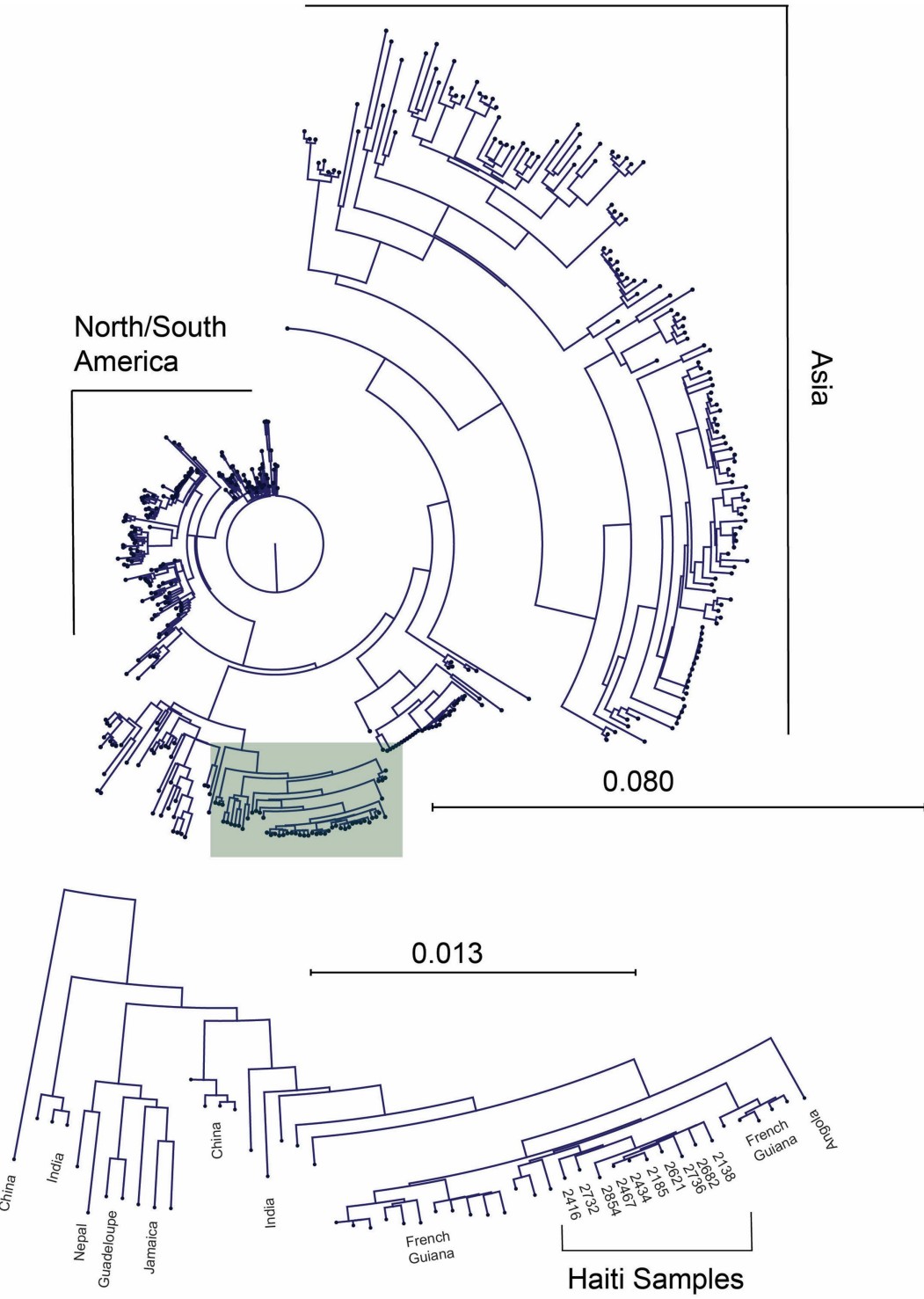

**Fig 2. Sequence analysis of the DENV-3 sequences from Haiti.** The near-complete DENV-3 sequences (n = 10) were aligned with 431 DENV-3 sequences from around the world. A phylogenetic tree (Jukes-Cantor, Neighbor Joining) was generated based off this alignment, showing a North/South America cluster, and Asian cluster, and a mixed region cluster. The sequences from Haiti (insert) group with this mixed region cluster.

the used RDTs. The PAHO PLISA Health Information Platform reported 6,298 and 3,316 dengue cases within Haiti during 2021 and 2022, respectively (https://www.paho.org/es/arbo-portal/dengue-datos-analisis/dengue-analisis-por-pais). Within the Dominican Republic, 3,746, 10,784, and 27,972 dengue cases were reported in 2021, 2022, and 2023, respectively, with DENV-2 described as the circulating serotype in 2022. Similarly, Louis et al. reported that 67% of school children (n = 61) from Fond-Parisien showed evidence of DENV-2 infections in 2021 [14]. These findings confirm that DENV was circulating both within Haiti and throughout Hispaniola between 2021–2022. Factors leading to this discrepancy of DENV detection instances may include spatiotemporal variations of DENV transmission patterns that vary by year, or relate to TNA product quality, as discussed below.

Evidence suggests that the extraction process of viral material from arboviral RDT strips results in a loss of viral RNA product [19–21]. This loss of viral genetic material has implications for this study, in that arbovirus infections may have gone undetected within our samples due to the decreased viral RNA amounts in the extracted product. However, it is worth noting that arboviral titers have been reported to reach magnitudes of $10^6$-$10^9$ copies or PFU/mL for symptomatic arboviral cases [26–28]. Given that the samples in this study were obtained from patients presenting to health care centers with symptoms, which may have consequently had higher viral titers during their symptomatic phase, the impact on detection from losses in viral RNA quantity may be minimal within the clinical context, but still represents a limitation in our study's ability to assess the burden of circulating arboviruses within febrile patients in Haiti.

The effects of storage temperature and duration on the stability of viral genetic material present on RDTs are still being investigated, but our findings suggest the success of this approach is time and storage condition dependent. One study found that while DENV was detectable from RDTs stored at -80°C, 4°C, and 35°C for up to 2 months, instances in which the RDT contained a lower titer of virus present within the RDT led to PCR-negative results, especially for RDTs stored at 35°C [19], indicating that viral genetic material may undergo degradation within RDTs when stored at room temperature. Another study demonstrated that sufficient nucleic acid quality was present within Ag-RDTs despite having been stored for up to 31 days at room temperature, allowing for the sequencing, serotyping and genotyping of DENV [21]. In this study, we extend the known viability of viral genetic material present on RDTs stored at room temperature for up to 15 months, with TNA product of sufficient quality for sequencing shown for up to 13 months (S1 Table).

The absence of other flaviviruses and CHIKV is in agreement with the PAHO PLISA Health Information Platform data, which states that the last CHIKV and ZIKV cases reported on Hispaniola were in 2019 (https://www.paho.org/en/arbo-portal/zika-data-and-analysis/zika-analysis-country). However, historically less than 4% of CHIKV, DENV and ZIKV reports from Haiti to PAHO have been laboratory-based, given a reliance on symptomatic diagnoses, particularly in rural areas, and thus the PAHO data may represent an underestimation of CHIKV and ZIKV circulation within the country [29]. Indeed, the study by Louis and colleagues suggests that these two arboviruses may still be present within the country, possibly at very low levels compared to DENV [14]. The lack of OROV detections within this study aligns with what is known of the OROV disease outbreak, which began in late 2023 within South America but has since been detected in the Caribbean islands of Cuba (May 2024) and Barbados (December 2024) [30]. Our team has acquired more recent RDT samples from 2024, with plans to test for OROV to continue assessing for the spread of this outbreak into Haiti.

The malaria RDTs used in this study were stored at room temperature, and for some samples these storage conditions lasted for more than 2 years. Given that previous studies have demonstrated reduced viral RNA recovery from RDTs stored at room temperature for extended periods of time, along with the observed loss of viral RNA upon extraction from RDTs, there may be instances of missed arbovirus case detections within this study, especially if the patient had a low-level viremia. This may have prevented detections of ZIKV cases within our study, given that ZIKV viremia can reach very low levels and that ZIKV viremia has a shorter duration that has been shown to decline once symptoms begin [31,26]. This may have also impacted arbovirus detection in RDT-positive samples, given that *Plasmodium* infections have been shown to reduce viral loads for arboviruses such as CHIKV [32]. Contributions from these two factors may explain why none of the samples in 2021 or 2022 were positive for any arboviruses. Another limitation of this study was the lack of

patient information, given that all RDTs were passively collected from healthcare centers. This prevents us from commenting on any possible demographic or epidemiological risk factors for DENV infections within Haiti. Given our sample analysis pipeline and the limited TNA eluate for each sample, we did not conduct testing for alphaviruses or Oropouche virus on flavivirus-positive samples. This may have resulted in missed detections of mixed infection within our samples. Additionally, while our testing strategy for flavivirus-positive samples was informed by the preliminary Sanger sequencing results demonstrating exclusively DENV-3, our subsequent decision to focus on identifying DENV in these samples may have led to overlooked instances of ZIKV co-infections, especially given that Sanger sequencing tends to favor the most abundant template present in the sample.

## Conclusions

As political instability impacts public health services within Haiti, it is essential for arboviral disease monitoring to continue in some shape or form. Viral genetic material present on discarded malarial RDTs from febrile patients shows promise for allowing detailed, molecular-based arbovirus characterization. These types of analyses allowed us to identify endemic circulation of DENV-3 throughout the Sud department of Haiti during June-September of 2023, with a 2023 sample positivity rate of 5.37% and instances of co-infections between DENV-3 and *P. falciparum*. We were able to use these types of samples to obtain near full sequences of the circulating DENV-3, adding to what is known regarding the genetic characteristics of DENV within the Americas. While using RDTs in this fashion to screen for arboviruses has limited epidemiological value and requires both coordination with local health care centers for sample collection and proper biosafety precautions, it remains a valuable tool for characterizing viruses in febrile patients, especially in areas where traditional study enrollment and sample collection are unfeasible due to resource limitations or wider security concerns.

## Supporting information

**S1 Table. Collection months and age of RDTs at time of extraction.** Sample sets in which a sample was determined positive by the pan-flavivirus assay and/or DENV assay are in bold.
(DOCX)

**S1 Fig. Metadata and testing results of RDT samples.**
(XLSX)

## Acknowledgments

The use of either trade or manufacturers' names in this report does not constitute an official endorsement of any commercial products.

The opinions, interpretations, conclusions, recommendations and views in this publication are those of the authors and do not necessarily reflect the official policy or position of the Uniformed Services University of the Health Sciences, Department of the Army, Department of Defense, nor the U. S. Government.

## Author contributions

**Conceptualization:** Bernard Okech, Keersten Ricks, Ian Sutherland, James Dunford, Jacques Boncy, Michael E. von Fricken.

**Data curation:** Graham A Matulis, Haley P Smith, Jacques Boncy.

**Formal analysis:** Ian Pshea-Smith, Abigail A Lilak, Christopher P Stefan, Jeffrey W Koehler.

**Funding acquisition:** Bernard Okech, Michael E. von Fricken, Jeffrey W Koehler.

**Investigation:** Graham A Matulis, Haley P Smith, Grant Hall, Rachel S Katich, Jennifer Williams, Shawn Hirsch, Raina Kumar, Keersten Ricks, Alexandre Existe.

**Methodology:** Korey L Delp, Christina E Douglas, Jeffrey W Koehler.

**Project administration:** Bernard Okech, Alexandre Existe, Jacques Boncy, Michael E. von Fricken, Jeffrey W Koehler.

**Software:** Ian Pshea-Smith.

**Supervision:** Keersten Ricks, Jeffrey R Kugelman, Ian Sutherland, James Dunford, Michael E. von Fricken.

**Visualization:** Graham A Matulis, Grant Hall, Ian Pshea-Smith.

**Writing – original draft:** Graham A Matulis, Grant Hall, Ian Pshea-Smith, Abigail A Lilak, Bernard Okech, Alexandre Existe, Michael E. von Fricken, Jeffrey W Koehler.

**Writing – review & editing:** Graham A Matulis, Ian Sutherland, James Dunford, Jacques Boncy, Michael E. von Fricken, Jeffrey W Koehler.

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
