## [Decision Letter · Decision Letter 0]

10 Dec 2025

Characterization of arbovirus infections in patients within Haiti by screening discarded malaria rapid diagnostic test cassettes

Dear Dr. Michael E. von Fricken,

Thank you for submitting your manuscript to PLOS Neglected Tropical Diseases. After careful consideration, we feel that it has merit but does not fully meet PLOS Neglected Tropical Diseases's publication criteria as it currently stands. Therefore, we invite you to submit a revised version of the manuscript that addresses the points raised during the review process.

Please submit your revised manuscript within by Feb 08 2026 11:59PM. If you will need more time than this to complete your revisions, please reply to this message or contact the journal office at plosntds@plos.org. Please include the following items when submitting your revised manuscript:

We look forward to receiving your revised manuscript.

Kind regards,

Takeshi Kurosu, D.V.M., Ph.D.

Academic Editor

David Safronetz

Section Editor

Shaden Kamhawi

co-Editor-in-Chief

Paul Brindley

co-Editor-in-Chief

**Additional Editor Comments :**

Although the authors’ study includes some interesting aspects, there are safety concerns regarding the reuse of RDT kits that have already been used for malaria testing or other purposes, such as dengue diagnosis. This point should be clearly stated, as suggested by one of the reviewers. In addition, many parts of the Methods section are inadequately described. The authors should therefore provide more detailed methodological information and add appropriate references. Finally, the authors should respond point by point to each reviewer’s comment.

**Journal Requirements:**

At this stage, the following Authors/Authors require contributions: Graham A Matulis, Haley P Smith, Grant Hall, Rachel S Katich, Korey L Delp, Christina E Douglas, Jennifer Williams, Shawn Hirsch, Raina Kumar, Ian Pshea-Smith, Abigail A Lilak, Bernard A Okech, Keersten Ricks, Christopher P Stefan, Alexandre Existe, Jeffrey R Kugelman, Ian Sutherland, James Dunford, Jacques Boncy, Michael E. von Fricken, and Jeffrey W Koehler. Please ensure that the full contributions of each author are acknowledged in the "Add/Edit/Remove Authors" section of our submission form.

Potential Copyright Issues:

i) Figure 1. Please (a) provide a direct link to the base layer of the map (i.e., the country or region border shape) and ensure this is also included in the figure legend; and (b) provide a link to the terms of use / license information for the base layer image or shapefile. We cannot publish proprietary or copyrighted maps (e.g. Google Maps, Mapquest) and the terms of use for your map base layer must be compatible with our CC BY 4.0 license.

5) In the online submission form, you indicated that "All other data is available upon reasonable request." All PLOS journals now require all data underlying the findings described in their manuscript to be freely available to other researchers, either

1. In a public repository

2. Within the manuscript itself

3. Uploaded as supplementary information.

6) Please ensure that the funders and grant numbers match between the Financial Disclosure field and the Funding Information tab in your submission form. Note that the funders must be provided in the same order in both places as well.

7) Please revise your current Competing Interest statement to the standard "The authors have declared that no competing interests exist.".

**Reviewers' Comments:**

**Comments to the Authors:**

**Please note that one review is uploaded as an attachment.**

Reviewer's Responses to Questions

**Key Review Criteria Required for Acceptance?**

**Methods**

-Are the objectives of the study clearly articulated with a clear testable hypothesis stated?

-Is the study design appropriate to address the stated objectives?

-Is the population clearly described and appropriate for the hypothesis being tested?

-Is the sample size sufficient to ensure adequate power to address the hypothesis being tested?

-Were correct statistical analysis used to support conclusions?

-Are there concerns about ethical or regulatory requirements being met?

Reviewer #1: The objectives of the study are clearly stated, and the manuscript presents a well-defined research question. However, the current version does not include sufficient details regarding the statistical analyses performed to support the conclusions. In addition, there is no explicit description of ethical considerations or regulatory compliance related to the use of clinical specimens. These aspects should be addressed in the revised manuscript to ensure methodological rigor and adherence to ethical standards.

Reviewer #2: Samples:

How was it decided what samples would be selected? Were all RDTs from the locations indicated used or was it a subset? If the latter, the authors need to describe why the subset is representative of the sites chosen.

RDT kits used:

Were the RDT all from the same manufacturer or was there variability? The authors should clarify this and if different kits used how this might impact results. This information should be included in the methods.

RDT storage conditions:

The storage conditions should be included in the methods. While it is stated in the conclusion that the samples were stored at room temperature it is unclear what this means as this could potentially vary widely. For example, the room may be air conditioned during the day only, or perhaps the room is open air in which room temperature would be expected to be mostly above >27�C in Haiti. It is also unclear how long the samples were stored in Haiti vs Florida.

Anonymized data:

Were names or laboratory numbers included on the RDT? The authors should include more information regarding how the RDT were anonymized to the investigators.

PCR conditions:

It is unclear how the PCR conditions were decided on. If the RNase P amplification was based off of a previous study then then it should be cited. If it was developed in house the authors should provide validation information. For the flavivirus PCR the authors should indicate how the assay was modified and if the modification is expected to lead to different results from the study cited. Are the alphavirus and OROV PCR conditions based off a previous study (if so, cite) or were they developed in house (if so provide validation information).

PCR interpretation:

The authors must elaborate on what an “appropriate PCR curve” is and also indicate what characteristics make a sample indeterminate.

Whole genome sequencing:

How were the published sequences chosen to include for phylogenetic analysis and where were the sequences retrieved from?

**Results**

-Does the analysis presented match the analysis plan?

-Are the results clearly and completely presented?

-Are the figures (Tables, Images) of sufficient quality for clarity?

Reviewer #1: 1. The manuscript uses RNase P as an indicator of nucleic acid integrity, which is appropriate. However, the interpretation of RNA stability would be more robust if pathogen detection levels were normalized to RNase P Ct values. This would allow the authors to account for sample-to-sample variability and better assess the relationship between RNA degradation and pathogen detection sensitivity. Without such normalization, comparisons across years or seasons are not feasible.

2. The authors infer the circulation of DENV-3 based on Sanger sequencing of flavivirus-positive samples. However, Sanger sequencing relies on PCR amplification, which tends to favor the most abundant template. Therefore, in cases of mixed infections or low viral load, minority pathogens may not be detected. Please discuss this limitation in the manuscript and consider whether alternative approaches (e.g., deep sequencing or multiplex PCR) could mitigate this bias.

3. The authors tested Alphaviruses and Oropouche virus only in samples that were negative for flaviviruses. However, mixed infections involving flaviviruses and other arboviruses have been reported. Therefore, testing should ideally include all samples regardless of flavivirus PCR results. Please consider expanding the screening strategy or clearly acknowledge this limitation in the discussion.

Reviewer #2: The results are for the most part clearly presented. However, Figure 2 could benefit from a more logical layout in which it is more obvious that the blown up phylogenetic tree is coming from the larger phylogenetic tree. It would also be beneficial to label the sequences shown in this study as “Haiti”. It is not clear or some clades if the country designation is indicating a tip or a clade (e.g., China, India, French Guiana).

There were 73 flavivirus positive/indeterminate samples and 68 identified as DENV-3. Were the 5 samples not identified as DENV-3 indeterminate? The authors should provide a statement regarding these 5 samples (and the interpretation of an indeterminate should be described in the methods as described above).

How were the subset of samples decided to run for alphavirus PCR, OROV PCR, and DENV WGS? Could there be bias?

**Conclusions**

-Are the conclusions supported by the data presented?

-Are the limitations of analysis clearly described?

-Do the authors discuss how these data can be helpful to advance our understanding of the topic under study?

-Is public health relevance addressed?

Reviewer #1: In the Discussion section (Lines 310–314), the authors argue that RNA loss during extraction may have minimal impact because symptomatic patients typically have high viral titers. However, this assumption overlooks scenarios such as mixed infections (e.g., malaria co-infection), where viral copy numbers for flaviviruses or alphaviruses could be significantly reduced. This could lead to false negatives and underestimation of co-circulating pathogens. Please address this limitation in the discussion and consider whether additional normalization or alternative approaches could improve detection accuracy.

Reviewer #2: Two minor comments for the discussion:

1. First sentence. The authors should explicitly state that the positivity rate is for symptomatic people testing for malaria to avoid any confusion with interpretation of positivity rate.

2. ZIKV typically has a much lower viremia than CHIKV and DENV. The authors should mention this in the discussion as it is likely that ZIKV could be missed using the RDT approach as it can even be difficult to identify in an infected person (low viremia and short duration of viremia).

**Editorial and Data Presentation Modifications?**

Reviewer #1: 1. Materials & Methods: Please revise the Materials and Methods section to ensure that all experimental products include complete manufacturer information, including company name, city, and country, in a consistent format.

2. Line 214: Please clarify the origin of the Plasmodium spp. positive results by PCR. Were these obtained from the current study or based on previously published data? In addition, the manuscript should include a discussion on why 36 samples that were negative by RDT tested positive by PCR. This discrepancy warrants explanation and could provide important insights into diagnostic sensitivity and sample integrity.

Reviewer #2: (No Response)

**Summary and General Comments**

Reviewer #1: The authors present an innovative approach for arbovirus surveillance in Haiti by repurposing discarded malaria rapid diagnostic tests (RDTs) as a source of nucleic acids for molecular analysis. The study collected 1,884 malaria RDTs from healthcare centers in Haiti’s Sud and Ouest Departments between 2021 and 2023. 73 samples were flavivirus-positive or indeterminate, and 68 were confirmed as dengue virus serotype 3 (DENV-3), all from 2023. NGS revealed high genetic similarity (>99.5%) between Haitian DENV-3 strains and those circulating in the Caribbean during 2022–2023. The authors conclude that discarded malaria RDTs can serve as a valuable tool for remote arbovirus surveillance and viral characterization, particularly in resource-limited or insecure settings.

While the manuscript demonstrates an innovative approach, it is important to note that the scenario described—where large numbers of used diagnostic kits are discarded and available for reuse—is not typical in most healthcare settings. Diagnostic kits that have come into contact with clinical specimens pose an infection risk and, under standard protocols, must be sterilized (e.g., by autoclaving) and disposed of as medical waste after use. For this reason, the authors should avoid language suggesting that this method could be generalized or routinely implemented. Instead, the discussion should clearly frame this approach as context-specific and acknowledge the biosafety and regulatory considerations involved.

Furthermore, the manuscript uses RNase P as an indicator of nucleic acid integrity, which is appropriate. However, the interpretation of RNA stability would be more robust if pathogen detection levels were normalized to RNase P Ct values. This would allow the authors to account for sample-to-sample variability and better assess the relationship between RNA degradation and pathogen detection sensitivity. Without such normalization, comparisons across years or seasons are not feasible. Please consider addressing this point.

Therefore, I recommend that this manuscript be considered for publication on PNTD only after major revision, addressing the points outlined below.

Reviewer #2: The manuscript by Matulis at al is well written and provides highly useful arbovirus surveillance information through means compatible with the current sociopolitical conditions in Haiti. While this is very well laid out the authors should ensure that the comments provided above are adequately addressed. Please note that on page 7, reference 22 should be reference 23 and many (all?) references after are impacted and should be adjusted as needed.

PLOS authors have the option to publish the peer review history of their article (what does this mean? ). If published, this will include your full peer review and any attached files.

**Do you want your identity to be public for this peer review?** For information about this choice, including consent withdrawal, please see our Privacy Policy .

Reviewer #1: No

Reviewer #2: No

**Figure resubmission:**
---

## [Decision Letter · Decision Letter 1]

25 Feb 2026

Dear Dr. von Fricken,

We are pleased to inform you that your manuscript 'Characterization of arbovirus infections in patients within Haiti by screening discarded malaria rapid diagnostic test cassettes ' has been provisionally accepted for publication in PLOS Neglected Tropical Diseases.

Best regards,

Takeshi Kurosu, D.V.M., Ph.D.

Academic Editor

David Safronetz

Section Editor

Shaden Kamhawi

co-Editor-in-Chief

Paul Brindley

co-Editor-in-Chief

The authors have satisfactorily responded to the revision requests, and the manuscript is considered acceptable for publication.

Reviewer's Responses to Questions

**Key Review Criteria Required for Acceptance?**

**Methods**

-Are the objectives of the study clearly articulated with a clear testable hypothesis stated?

-Is the study design appropriate to address the stated objectives?

-Is the population clearly described and appropriate for the hypothesis being tested?

-Is the sample size sufficient to ensure adequate power to address the hypothesis being tested?

-Were correct statistical analysis used to support conclusions?

-Are there concerns about ethical or regulatory requirements being met?

Reviewer #1: The authors have adequately addressed the major and minor comments from Reviewer 1 and the Editor. The revisions improved clarity, safety considerations, and methodological transparency.

**Results**

-Does the analysis presented match the analysis plan?

-Are the results clearly and completely presented?

-Are the figures (Tables, Images) of sufficient quality for clarity?

Reviewer #1: • Biosafety and context-specific use of discarded RDTs are now clearly explained.

• Methods were expanded with additional details, including reagents, storage conditions, and assay references.

• Limitations regarding mixed infections, RNA degradation, and detection bias were added.

• Ethical approvals and anonymization procedures are now properly documented.

**Conclusions**

-Are the conclusions supported by the data presented?

-Are the limitations of analysis clearly described?

-Do the authors discuss how these data can be helpful to advance our understanding of the topic under study?

-Is public health relevance addressed?

Reviewer #1: The manuscript has satisfactorily addressed reviewer concerns and is suitable for acceptance.

**Editorial and Data Presentation Modifications?**

Reviewer #1: All requested editorial and minor data‑presentation improvements have been incorporated. No additional modifications are necessary, and the manuscript is suitable for acceptance.

**Summary and General Comments**

Reviewer #1: The revised manuscript presents a clear, well‑executed, and context‑appropriate study evaluating the use of discarded malaria RDTs as a source of nucleic acids for arbovirus surveillance in Haiti. The work is timely and significant, given ongoing constraints in public health infrastructure and limited capacity for conventional surveillance. The authors successfully demonstrate the feasibility of detecting and genetically characterizing DENV‑3 using RDT‑derived material, and they provide robust phylogenetic evidence linking the identified strains to recent Caribbean circulation. The study offers meaningful insights into surveillance strategies in settings facing security, logistical, and resource challenges.

The strengths of the work include:

• A large number of RDTs analyzed (n=1,886), providing valuable context‑specific surveillance coverage.

• Clear detection workflow with appropriate molecular methods, complemented by next‑generation sequencing.

PLOS authors have the option to publish the peer review history of their article (what does this mean? ). If published, this will include your full peer review and any attached files.

**Do you want your identity to be public for this peer review?** For information about this choice, including consent withdrawal, please see our Privacy Policy .

Reviewer #1: No

---

## [Editor Report · Acceptance letter]

Dear Dr. von Fricken,

We are delighted to inform you that your manuscript, "Characterization of arbovirus infections in patients within Haiti by screening discarded malaria rapid diagnostic test cassettes ," has been formally accepted for publication in PLOS Neglected Tropical Diseases.

Best regards,

Shaden Kamhawi

co-Editor-in-Chief

Paul Brindley

co-Editor-in-Chief
